# Is There a Role for Intraoperative Neuromonitoring in Intradural Extramedullary Spine Tumors? Results and Indications from an Institutional Series

**DOI:** 10.3390/jpm13071103

**Published:** 2023-07-06

**Authors:** Manuela D’Ercole, Quintino Giorgio D’Alessandris, Michele Di Domenico, Benedetta Burattini, Grazia Menna, Alessandro Izzo, Filippo Maria Polli, Giuseppe Maria Della Pepa, Alessandro Olivi, Nicola Montano

**Affiliations:** 1Department of Neurosurgery, Fondazione Policlinico Universitario A. Gemelli IRCCS, 00168 Rome, Italy; manuela.dercole@policlinicogemelli.it (M.D.); quintinogiorgio.dalessandris@policlinicogemelli.it (Q.G.D.); michele.didomenico@policlinicogemelli.it (M.D.D.); izzo.alessandro88@gmail.com (A.I.); filippomaria.polli@policlinicogemelli.it (F.M.P.); giuseppemaria.dellapepa@policlinicogemelli.it (G.M.D.P.); alessandro.olivi@policlinicogemelli.it (A.O.); 2Department of Neuroscience, Neurosurgery Section, Fondazione Policlinico Universitario A. Gemelli IRCCS, Università Cattolica del Sacro Cuore, 00168 Rome, Italy; benedetta.burattini@gmail.com (B.B.); mennagrazia@gmail.com (G.M.)

**Keywords:** spinal tumors, intradural extramedullary tumors, intraoperative neurophysiological monitoring

## Abstract

While intraoperative neurophysiological monitoring (IONM) is considered a standard for intramedullary spinal cord tumor surgery, the effective role of IONM in intradural extramedullary (IDEM) tumors is still debated. We present the results of 60 patients affected by IDEM tumors undergoing surgery with the aid of IONM. Each patient was evaluated according to the modified McCormick scale (MMS) at admission, discharge and at follow-up. During surgery, motor evoked potentials (MEPs) and somatosensory evoked potentials (SEPs) were studied using the Medtronic NIM-eclipse^®^ 32-channel system (Medtronic Xomed, Inc. 6743 Southpoint Drive North Jacksonville FL USA). Patients’ age, gender and tumor location did not affect MMS modifications. Tumors involving more than three levels had an increased likelihood of MMS worsening, while meningioma pathology was associated with worse preoperative and 1-year follow-up MMS. No MEP amplitude ratio was able to predict clinical variations, while intraoperative SEP worsening was associated with 100% risk of poor MMS at discharge and with 50% risk of poor MMS at long-term follow-up. In our opinion, SEP monitoring is a valid tool that may contribute to the preservation of the patient’s neurological status. MEP monitoring is not mandatory in IDEM surgery while more studies are required to explore the feasibility and the role of D-wave in this kind of surgery.

## 1. Introduction

Spinal tumors, either in their primitive or secondary origin, account for 10% of all central nervous system neoplasms. Among them, about 5% are intramedullary spinal cord tumors and 95% are extramedullary spinal cord tumors. Extramedullary tumors can be either extradural or intradural. While extradural lesions are commonly bone tumors, both primary and metastatic in their origin, the most common types of intradural–extramedullary (IDEM) tumors are meningiomas and nerve sheath tumors (schwannomas and neurofibromas), followed by metastases, dermoid tumors, teratomas, paragangliomas, ependymoma of the cauda equina or filum terminalis and hemangioblastomas [1]. IDEM tumors can present with a wide variety of symptoms, ranging between generic low back pain and more specific neurological signs related to compression of spinal cord and/or spinal roots. The diagnosis of IDEM tumors is made by spinal magnetic resonance imaging (MRI) with gadolinium administration which can demonstrate the relationship of the tumor with the dura mater and spinal cord and the presence or not of associated myelopathy. Although various strategies of treatment have been suggested according to the specific diagnostic hypothesis, surgical resection has a primary role in the case of incoming neurological deficits in order to provide a quick prevention of further deterioration, beyond the goal of a maximal safe resection related to the oncological result. The surgical removal of tumor is achieved through the opening of posterior corresponding bone and ligamentous structures. Laminoplasty is currently preferred to classic laminectomy, with the clear advantage of preventing deformity onset, related to the shift of the weight-bearing axis forward and a consequent increase in the force on the anterior vertebral body, usually seen in laminectomy. [2]. However, surgery carries a significant risk of intraoperative morbidity, ranging from 3.7% to 7.5% [3,4]. Intraoperative neurophysiological monitoring (IONM) allows for a real-time check of the functional integrity of both the spinal cord and the nerve roots. The importance of IONM should be considered not only in the middle of the surgical procedure, i.e., the tumor removal, but throughout the entire surgical procedure, from positioning of the patient on the operating table to laminotomy, dura opening and final reconstruction (duroplasty and laminoplasty). Each critical change in IONM requires a prompt analysis of the most recent surgical and anesthesiologic steps. After a check of vital parameters, with specific regard to blood pressure and body temperature, any changes in anesthesia schedule (such as addition of volatile anesthetics) should be verified. The subsequent step is a careful inspection of surgical field in order to play out the opportune corrective action, usually consisting in modifying traction on the tumor or retraction on the surrounding spinal cord parenchyma, optimizing hemostasis, preferably with warm saline solution irrigation, application of a sponge, cottonoid or thrombin-coated pad, instead of bipolar coagulation, and applying nimodipine in cases of visible vasospasm. In addition to prevention of neurologic iatrogenic injury, IONM may also affect the oncological prognosis by conditioning the extent of surgical excision (for example, if there is a decrease in neurophysiological responses the neurosurgeon can decide to stop the tumor removal) and thus the eventual tumor regrowth. Application of IONM in patients affected by some degree of neurological impairment before surgery should be evaluated case by case according to their specific degree of monitorability, although patients with the lowest levels of neurological reserves, usually the least monitorable, are just those that would obtain the greatest benefit from IONM. Surprisingly, Korn et al. found that patients with more severe preoperative McCormick grading (3–4) were more likely to have stable intraoperative monitoring when compared with patients with milder grading. [5]. Although there is increasing application of IONM in surgery for IDEM tumors [6], its effective role is still debated, with particular concern about its role in modifying the surgical strategy and predicting the postoperative neurological function. The aim of our study was to report the effectiveness of IONM in a series of patients submitted to surgery for IDEM tumors.

## 2. Materials and Methods

### 2.1. Patients’ Enrollment and Data Collection

We enrolled 60 patients who underwent surgery for IDEM tumors at the Department of Neurosurgery, Fondazione Policlinico Gemelli IRCCS, Rome, Italy, between January 2018 and December 2021. All patients signed an informed consent form complying with the institutional research committee of Fondazione Policlinico Gemelli IRCCS. Baseline patients’ data are presented in Table 1. Tumor location was classified as cervical, upper thoracic (T1–T6), lower thoracic (T7–conus) and lumbosacral. Muscle strength was graded using the modified McCormick scale (MMS) and dichotomized as good (MMS 1–2) or poor (MMS 3–5). MMS was recorded at 3 time-points, namely admission, discharge and last follow-up of ≥1 year. Variations in MMS at discharge and at the end of follow-up, as compared to baseline, were also recorded. Four patients were lost at follow-up after discharge while one patient harboring a malignant peripheral nerve sheath tumor died 5 months after surgery due to systemic progression of the disease.

### 2.2. Intraoperative Monitoring

Motor evoked potentials (MEPs) and somatosensory evoked potentials (SEPs) were studied using the Medtronic NIM-eclipse^®^ 32-channel system. IONMs were recorded during the whole procedure from patient positioning to laminoplasty and wound closure. Our anesthesiologic plan includes a standard value of mean arterial pressure above 80 mmHg, in order to ensure adequate blood supply to the spinal cord during the surgical procedure. In case of changes in IONM data the first due check is related to any changes in anesthesiologic plan: every administration of curare or anesthetic gas requires an adequate disposal period. We usually acquire IONM data before and after pronating the patients, in order to identify any changes and eventually adjust the position. All the patients underwent laminotomy and subsequent laminoplasty, sparing the articular joints in order to preserve the spinal stability and avoid instrumentation. The procedures were performed by four surgeons of the same team. IONM changes can occur during surgical dissection (Figure 1); in these cases, the surgeon usually proceeds to irrigate with saline solution, modify the surgical strategy and, in some cases, systemically administer steroids. We have a specific protocol about intraoperative MAPs.

#### 2.2.1. Motor Evoked Potentials

MEPs were elicited using electric transcranial stimulation (NIM-eclipse^®^ 32-channel system, Medtronic Xomed, Inc. 6743 Southpoint Drive North Jacksonville FL USA). Corkscrew electrodes were placed in C1–C2 positions (10–20 international EEG system) using C1 or C2 as anode for right or left limbs, respectively. Series of 5–7 stimuli (pulse width 75–500 µs, 250–500 Hz, 75–750 V) were delivered in order to obtain at least a stable muscular response for each studied limb with 100 µV amplitude [7]. Muscular responses were recorded from biceps brachii, wrist extensor and abductor pollicis brevis for upper limbs; quadriceps, tibialis anterior and abductor hallucis for lower ones. For lesions located in conus or cauda equina, sural and external anal sphincter were also studied. MEPs were recorded regularly about every 5 min. During tumor resection the interval was reduced to 1–2 min.

#### 2.2.2. Somatosensory Evoked Potentials

SEPs were elicited using surface electrodes placed on the median nerve at the wrist for upper limbs and on the tibialis posterior nerve at the ankle for lower limbs. Recording was performed from the scalp using corkscrew electrodes placed on C3′, C4′ and Cz’ referring to Fz (10–20 international EEG system). For baseline definition, stimulation intensity was adjusted in order to obtain stable and reliable responses with maximal amplitude (single pulse, pulse width 300 µs, 4.1 Hz, 10–50 mA). A 50% amplitude decrease and/or 10% latency increase were considered significant variations [8].

#### 2.2.3. MEP Amplitude Ratios

As already described [9], raw MEP amplitudes were elaborated and the following amplitude ratios were calculated: minimum-to-baseline amplitude ratio (MBR, the ratio between lowest and baseline MEP amplitude); final-to-baseline amplitude ratio (FBR, the ratio between final and baseline MEP amplitude); and recovery value (RV, FBR minus MBR). Amplitude ratios were obtained from each monitored muscle. For each patient, we then calculated the best value, the worst value, the mean value and the median value for each index and correlated these values with clinical parameters.

#### 2.2.4. Statistical Analysis

Continuous variables were reported as mean values ± standard deviation (SD) or median values with range. Categorical variables were reported as absolute and relative frequencies. Comparison of categorical variables was conducted with the chi-square statistics, using the Fisher exact test when appropriate. Comparison of continuous variables between groups was conducted using the Mann–Whitney U test (2 groups) or the Kruskal–Wallis test (3 groups). An a priori power analysis on comparison of MEP amplitude ratios between patients with good vs. poor 1-year outcome was performed. A normal distribution of the variable had to be assumed; a 50% difference in means between the two groups was postulated, with a 25% standard deviation and a ratio of poor/good outcome patients of 0.1. Considering a cohort of 55 patients, a power of 97.5% was achieved, with a type I error probability set at 5%. To analyze the accuracy of amplitude indices in predicting postoperative MMS, receiver operating characteristic (ROC) curves were plotted and the area under the curve was calculated. The best cut-off was defined as the value at which the Youden index (i.e., the difference between sensitivity and 1-specificity), had its maximum value. The a priori power analysis of the ROC curve, considering a population of 55 patients with 1-year follow-up and a ratio of poor/good outcome of 0.1, showed a suboptimal power (65%) of demonstrating a 0.8 area under the curve of the studied parameter, with a type I error probability set at 5%. A two-sided *p*-value < 0.05 was considered significant. Data were analyzed using MedCalc ver 20.218 (MedCalc Software Ltd., Osted, Belgium).

## 3. Results

The tumor findings are reported in Table 1. Briefly, the most frequent tumor subtypes were meningiomas and schwannomas as expected. Among 28 schwannomas, 16 were located in lumbosacral spine, 9 in the thoracic spine and 3 in the cervical spine, respectively. Among 19 meningiomas, 12 showed an anterolateral attachment and the remaining 7 were dorsally located. In all the patients a gross total resection was obtained, as assessed by postoperative MRI, routinely performed within 48–72 h after surgery. In all 19 cases of meningioma a Simpson grade II resection was obtained, with dural preservation, thus allowing an easier layer reconstruction. The mean surgical time was 3.19 ± 1.13 h.

### 3.1. Predictors of Preoperative MMS

Median preoperative MMS was 2 (Table 2). A good preoperative MMS was recorded in 47 patients (78.3%) and a poor preoperative MMS in 13 patients (21.7%) (Table 2). Preoperative MMS was not influenced by age (<65 years vs. ≥65 years, *p* = 0.56, Fisher exact test), gender (*p* = 0.72, Fisher exact test), tumor location (*p* = 0.96, chi-square test) and number of involved levels (*p* = 0.89, chi-square test). Poor preoperative MMS was more frequently recorded in patients affected by meningioma as compared with schwannoma (36.8% vs. 7.1%, respectively; *p* = 0.0120, Fisher exact test).

### 3.2. MEP Amplitude Ratios

Table 3 summarizes the studied amplitude ratios. We observed a remarkable intraoperative variability in amplitude ratios, which is partially due to fluctuations in the level of sedation.

### 3.3. Predictors of MMS at Discharge

Median postoperative MMS was 2. We observed an improvement in MMS value in eight cases (13.3%) and a worsening in nine patients (15%) (Table 2). Postoperative MMS was not correlated with age (*p* = 0.99, Fisher exact test), gender (*p* = 0.84, Fisher exact test), tumor location (*p* = 0.77, chi-square test), number of involved levels (*p* = 0.26, chi-square test), meningioma vs. schwannoma histology (*p* = 0.13, Fisher exact test). Patients’ age, gender and tumor location did not influence MMS variation at discharge as compared to baseline. Tumors involving more than three levels had an increased likelihood of MMS worsening at discharge (100% vs. 12.1% in those involving ≤3 levels; *p* = 0.0203, Fisher exact test). We categorized the tumors as smaller than 2 cm, between 2 and 5 cm and larger than 5 cm. There were no differences in terms of outcome between the first two groups; conversely, tumors larger than 5 cm had an increased likelihood of MMS worsening at discharge (*p* = 0.0352, Fisher exact test). No correlations were found between MEP amplitude ratios and MMS at discharge (Mann–Whitney U test). Similarly, no clinically sensible correlation between MEP amplitude ratios and MMS variation at discharge were found (Kruskal–Wallis test). Conversely, a significant correlation was found between intraoperative SEP variations and MMS at discharge. Intraoperative SEP worsening was observed in no patients with good postoperative MMS vs. 33.3% patients with poor postoperative MMS (*p* = 0.0006, Fisher exact test; Table 4).

### 3.4. Predictors of MMS at the End of Follow-up

Long-term follow-up (≥1 year) was available for 55 patients (see Materials and Methods). Median MMS at the end of follow-up was 2. Notably, a good outcome was recorded in 90.9% of patients (Table 2). In analogy with the other time-points, final outcome was not correlated with age (*p* = 0.65, Fisher exact test), gender (*p* = 0.43, Fisher exact test) and tumor location (*p* = 0.39, chi-square test). Number of involved levels and a tumor size larger than 5 cm were negatively correlated with final outcome (*p* = 0.0123 and *p* = 0.0205, respectively, chi-square test). Meningioma pathology confirmed its negative value for final outcome (23.5% poor outcome vs. 3.7% in schwannoma, *p* = 0.0461, Fisher exact test). Number of involved levels was also negatively correlated with MMS variation as compared with baseline (*p* = 0.0001, chi-square test).

A trend of correlation was found between MMS at 1 year and best MBR (*p* = 0.0946, Mann–Whitney U test). No other correlations were found between MMS at 1 year and MEP amplitude ratios. In ROC analysis, best MBR was able to sub optimally predict final outcome, with an AUC of 0.718 and a best cut-off set at 94% (Figure 2). However, by adopting this cut-off, we found that all patients with best MBR >94% had a good long-term outcome vs. 80.2% of patients with best MBR < 94% (*p* = 0.0208, Fisher exact test; Figure 3). No MEP amplitude ratio was able to predict clinical MMS variations at the end of follow-up compared with preoperative baseline.

Again, intraoperative SEP variations were significantly associated with 1-year MMS. Intraoperative SEP worsening was observed in 4.3% patients with good 1-year MMS vs. 40% patients with poor 1-year MMS (*p* = 0.0433). In other words, intraoperative SEP worsening was associated with 100% risk of poor MMS at discharge and with 50% risk of poor MMS at long-term follow-up (Table 4).

**Table 4 jpm-13-01103-t004:** Intraoperative SEP variations and MMS.

Intraoperative SEP Variation	Good MMS at Discharge	*p*	Good MMS at Last Follow-Up	*p*
Stable	37/49	0.0006	44/47	0.0433
Worsened	0/6		2/4	

## 4. Discussion

The historical basis of IONM can be detected in Penfield and Boldrey’s research, published in 1937, describing a systematic mapping of the cerebral cortex [10]. These findings were exclusively applied in epilepsy surgery for almost half a century, until 1972, when Nash and his group proposed SEPs as markers of the functional integrity of the spinal cord during surgery [11]. The following experiences, however, revealed the lack of SEPs’ ability in monitoring the anterolateral column, usually involved in anterior spinal artery syndrome [12,13,14]. The subsequent introduction of combined use of MEPs and D-wave in the 1990s markedly improved IONM’s value and reliability. In the common experience, the loss of MEPs may not be significant in the presence of a D-wave preserved up to 50% of its baseline amplitude; the outcome will result in only transient motor deficits [15].

The indispensable role of IONM in surgery of intramedullary tumors has been widely demonstrated, although its prognostic significance for long-term outcome should still be definitively assessed. Since outcome could improve at long-term evaluation when compared to the immediate postoperative status, because of neuronal plasticity and rehabilitation, both the short-term and long-term neurologic status should be taken into account in evaluation of efficacy of IONM. In 2005, Quiñones-Hinojosa et al. found a strong correlation between the loss or the degree of MEP impairment and motor function at 15 months’ follow-up [16]. Similarly, Sala et al. reported in 2006 their experience in MEP monitoring in intramedullary tumors, showing a significant better long-term motor outcome in the IONM group compared with a non-IONM group [17]. Both cited papers, however, considered only MEP monitoring; definitive data on multimodal IONM are still lacking. Single-method IONM approaches have proven to be insufficient in assessing both the ascending and descending pathways. For this reason, the current IONM technique is a multimodal combination as a result of experience and expertise [18]. IONM modalities are chosen according to the specific anatomic structures involved; in cervical and thoracic procedures the integrity of sensory and motor pathways in the spinal cord is assessed through the combined monitoring of SEPs and MEPs. In lumbosacral procedures, integrity of nerve roots should be preserved, thus EMG monitoring is the focus of IONM. On the other hand, routine application in IDEM tumors is still debated.

At our institution, IONM is currently applied during each surgical procedure, either in cranial, spinal or peripheral districts, at risk for neurological function impairment. The lack of availability of this precious tool in our experience is mainly due to organizational issues; the IONM service (either devices or technical and medical staff) relates directly to neurosurgical departments. Routine application of IONMs allowed us to increase our experience and to reduce OR setup times.

Most published series are flawed due to small samples sizes [19,20] or short follow-up time [21]. Moreover, there is no agreement about the specific modality of IONM according to the anatomic level of IDEM tumors [22]. MEPs can show wide amplitude and morphological variability; the absence or the loss of responses is considered the most effective warning criterion, followed by changes in thresholds, in waveform or in amplitude. It has been reported that MEP sensitivity in anticipating postoperative motor deficits in spine surgery ranges from 75% to 100%, with a specificity from 25% to 100% [15,23], with the conus medullaris and cauda equina being the sites showing the highest specificity [24]. In our series no MEP amplitude ratio was able to predict clinical neurological variations at discharge and at the end of follow-up compared with preoperative baseline, and only the best MBR showed a tendency to suboptimally predict the final outcome with a cut-off of 94%. In other words, the preservation of MEP during the surgery can predict a good outcome at follow-up while the reduction of MEP does not necessarily predict a poor outcome in the long-term follow-up. SEPs provide monitoring of tactile pathways in the dorsal column and medial lemniscus through stimulation of the median nerve at the wrist, the posterior tibial nerve at the ankle and the pudendal nerve and through recording by corkscrew-like electrodes inserted in the scalp [25]. SEP sensitivity in spinal surgery ranges between 75% and 94%, while specificity from 50% to 100% in anticipating postoperative deficits. A 50% drop in amplitude and/or a 10% prolongation in latency are considered as warning criteria [26]. The results of our series, although limited to a small sample of patients, show a high predictive value of SEPs for neurological function both at discharge that at the end of follow-up. In 2008, Sandalcioglu et al. reported on their experience of IONM with only SEPs on 131 spinal meningiomas; since neurological status at follow-up was improved or unchanged in more than 96% of patients, they concluded that IONM was not necessary to reach a good clinical outcome [27]. On the contrary, Forster et al., from their experience of 141 patients affected by IDEMs belonging to a larger series of spinal tumors, reported an important contribution from IONM, whose variations allowed changes in surgical strategy in a range between 5.67% and 17.7% of patients [3]. Other authors found no benefit provided by IONM in avoiding postoperative complications in surgical treatment of spinal nerve sheath tumors [28]. Our protocol of multimodal IONM in IDEM surgery does not include routine measurement of direct D-wave, whose actual role in IDEM tumor surgery is still debated, differently from its confirmed relevance in surgery for intramedullary tumors. The D-wave provides a direct monitoring of fast-conducting fibers in the corticospinal tract; as fibers numerically decrease craniocaudally, its use is limited in the cord up to T10–11. The D-wave is elicited by a single-pulse stimulus and is recorded from the epidural or subdural spaces in the exact midline of the spinal cord. A warning criterion is a decrease of more than 50% of the baseline amplitude [29]. Only a few studies have specifically addressed the issue of D-wave accuracy in IDEM tumor surgery [30]. Advantages of D-wave monitoring consist in its real-time continuous feedback on the integrity of the corticospinal tract, differently from muscle MEP response, recorded in an on–off fashion; moreover, D-wave deterioration usually occurs gradually, thus allowing corrective measures. At least in theory, the D-wave might find a useful application in monitoring the functional aspects of the spinal cord during its manipulation in surgery of IDEM tumors located in an anterolateral position. Ghadirpour et al., in a study conducted on 108 patients affected by IDEM, assessed the feasibility of D-wave monitoring. They found that the D-wave monitoring was feasible in all patients without severe preoperative motor deficits and that the D-waves could predict the onset of postoperative deficits in IDEM patients [31]. Finally, we found no correlation between the neurological outcome and age, gender, and tumor location. Conversely, the number of involved levels and meningioma pathology showed a strong correlation with worse neurological outcome. These data agree with what is reported in the literature, because a more extended tumor and the fact that meningiomas usually present an anterolateral dural attachment will probably lead to a higher manipulation of the spinal cord. It has been previously reported that the anterolateral location of tumor to the spinal cord is a risk factor associated with significant IONM changes [32] and that there is a worse neurological outcome in tumors located ventral to the spinal cord [33]. This may be the result of various factors, such as tumor proximity to the anterior spinal vasculature and the anterolateral corticospinal tracts, the greater difficulty in accessing anterior tumors and thus the degree of spinal cord manipulation as well as a less well-defined border between tumor and healthy spinal cord tissue.

## 5. Conclusions

Although the effective role of IONM in IDEM tumors is still a matter of debate, we found that SEP monitoring was a valid tool that, providing continuous feedback to the neurosurgeon, may contribute to the preservation of the patient’s neurological status. MEP monitoring is, in our opinion, not mandatory in IDEM surgery while more studies are required to explore the feasibility and the actual role of the D-wave in this kind of surgery. In conclusion, the correlations found between electrical changes and clinical outcomes provide some evidence that IONM’s employment may contribute to preservation of the patient’s neurological status, mainly in identifying neural injury at initial reversible stages, thus allowing applications of due corrective measures.

## Figures and Tables

**Figure 1 jpm-13-01103-f001:**
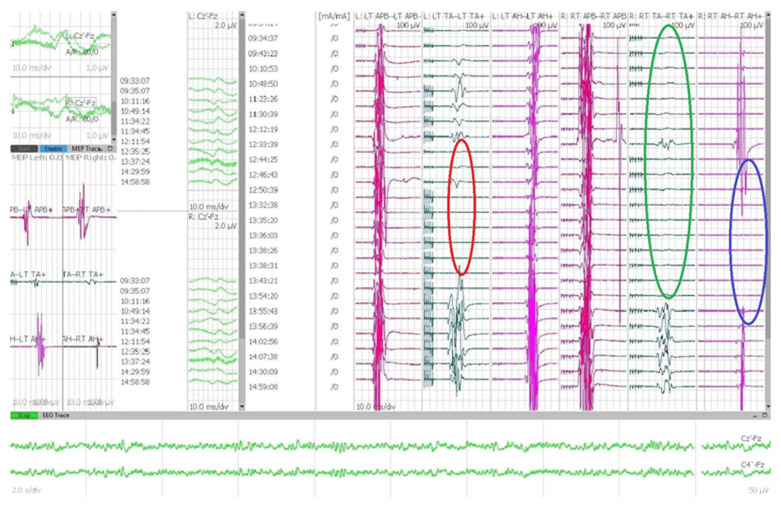
Example of IONM in a case of thoracic meningioma: red circle shows a temporary disappearance of muscular response from left tibialis anterior; green circle shows absence at baseline acquisition of the right tibialis anterior muscular response, followed by an intraoperative recovery; blue circle outlines a temporary drop in MEP from the right abductor halluces.

**Figure 2 jpm-13-01103-f002:**
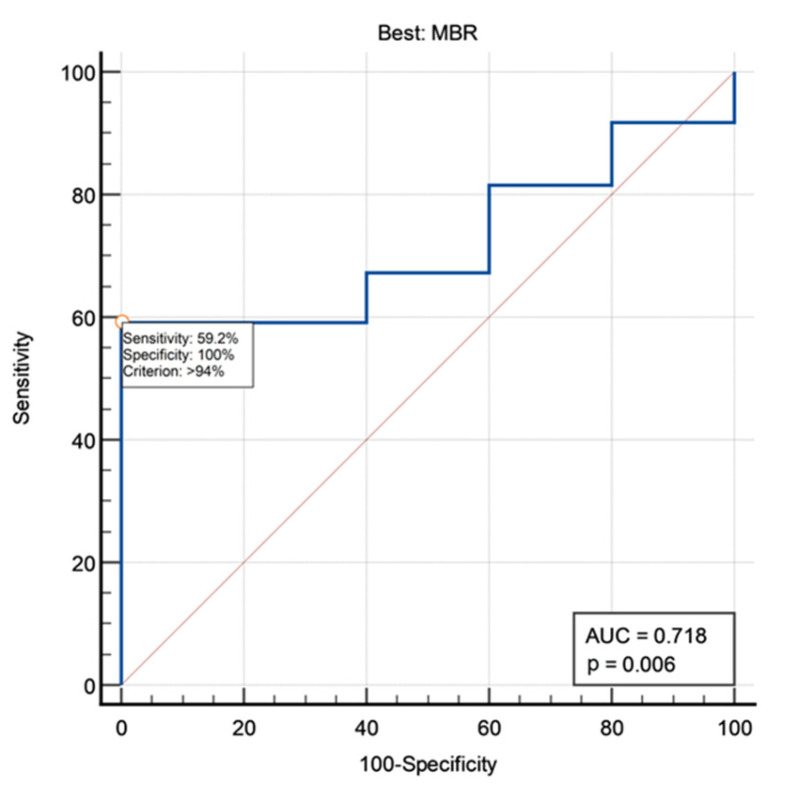
ROC curve analyzing the predictive value of best MBR for final outcome. MBR, minimum-to-baseline amplitude ratio.

**Figure 3 jpm-13-01103-f003:**
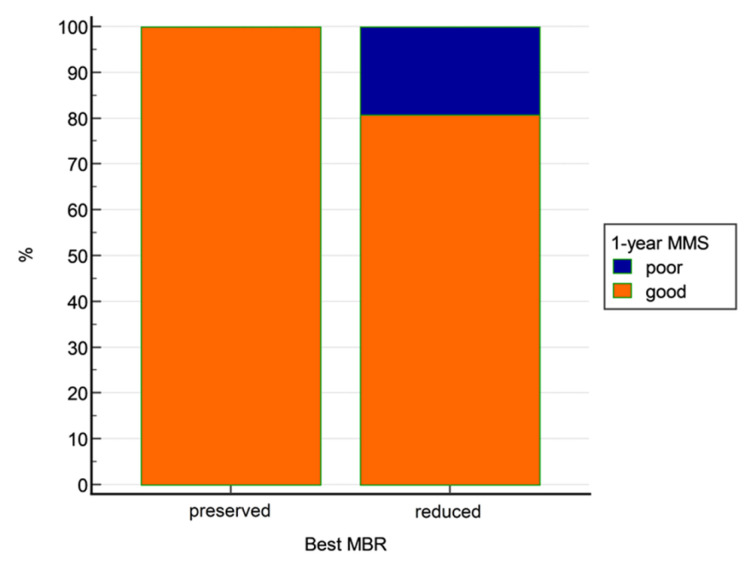
Outcome at 1-year follow-up depending on best MBR.

**Table 1 jpm-13-01103-t001:** Baseline Patient characteristics.

Feature	Value
Age (mean ± SD)	56.3 ± 15.9
Male:Female (%)	21:39 (35:65%)
Tumor location	
Cervical	12 (20%)
Upper thoracic	14 (23.3%)
Lower thoracic	14 (23.3%)
Lumbosacral	20 (33.3%)
Number of involved levels	
1	16 (26.7%)
2	36 (60%)
3	6 (10%)
4	1 (1.7%)
5	1 (1.7%)
Pathology	
Meningioma	19 (31.7%)
Schwannoma	28 (46.7%)
Paraganglioma	2 (3.3%)
Hemangiopericytoma	2 (3.3%)
Neurofibroma	1 (1.7%)
Other	8 (13.3%)

**Table 2 jpm-13-01103-t002:** Modified McCormick Scale of the included patients.

Modified McCormick Scale	Preoperative	Postoperative	At Follow-Up
Good	47 (78.3%)	41 (68.3%)	50 (90.9%)
Poor	13 (21.7%)	19 (31.7%)	5 (9.1%)
Improved *	NA	8 (13.3%)	17 (30.9%)
Stable *	NA	43 (71.7%)	36 (65.5%)
Worsened *	NA	9 (15%)	2 (3.6%)

* Compared to preoperative value. NA, not applicable.

**Table 3 jpm-13-01103-t003:** Intraoperative MEP amplitude ratios.

Amplitude Ratio	Mean	Median	SD	Minimum	Maximum
Mean FBR	234.8%	123.0%	371.8%	27.2%	2619.3%
Median FBR	113.3%	95.8%	64.8%	17.2%	476.5%
Best FBR	961.0%	197.9%	2225.9%	65.0%	14,674.2%
Worst FBR	49.6%	48.5%	33.2%	1.7%	131.6%
Mean MBR	58.1%	58.7%	21.0%	14.0%	96.6%
Median MBR	58.3%	62.6%	26.5%	5.1%	100.0%
Best MBR	88.6%	96.6%	15.6%	36.3%	100.0%
Worst MBR	25.7%	17.2%	22.2%	0.6%	76.0%
Mean RV	176.8%	55.9%	370.0%	2.6%	2541.8%
Median RV	52.8%	31.7%	65.7%	0.0%	465.2%
Best RV	890.4%	145.1%	2219.2%	11.7%	14,576.6%
Worst RV	9.5%	2.1%	19.2%	0.0%	120.9%

FBR, final-to-baseline amplitude ratio; MBR, minimum-to-baseline amplitude ratio; RV, recovery value.

## Data Availability

Source data are available from the corresponding author upon reasonable request.

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
