# Peer review of "Is There a Role for Intraoperative Neuromonitoring in Intradural Extramedullary Spine Tumors? Results and Indications from an Institutional Series"

_jpm, 2023, doi:10.3390/jpm13071103_

Round 1

Reviewer 1 Report

The authors present a retrospective study of 60 patients undergoing spine surgery for intradural extramedullary spine tumors to determine the importance of intraoperative monitoring. Tumors involving more than 3 levels had an increased likelihood of MMS worsening, while 21 meningioma pathology was associated with worse preoperative and 1-year follow-up MMS. No 22 MEPs amplitude ratio was able to predict clinical variations, while intraoperative SEPs worsening 23 was associated with 100% risk of poor MMS at discharge and with 50% risk of poor MMS at long- 24 term follow-up.  the authors concluded that IOM should be implemented and may predict outcomes.

The authors did a great job presenting their findings, however, I have the following comments that may improve the quality of their manuscript:

- did any of the patient received instrumentation? if so, how many levels? receiving instrumentation is crucial to adjust for. if no instrumentation was used, I encourage the authors to add it to the methods. 

- how many patients achieved total resection? what was the operative time? 

- How was a change in IOM approached? is there any protocol at the hospital or a list to run regarding things to check that may contribute to the changes, such as patient positioning?

- was a power analysis done? I would emphasize on that in the methods, as the results are clearly underpowered, especially the KM plots. 

- Please tone down your conclusions and remove "clear evidence"

- please discuss the limitations in more depth, including the low number of patients involved. 

- how many surgeons were involved? 

- what was the approach used? are they all posterior? was there any costotransversectomies? 

- Please add to the discussion a paragraph about how, at your institution, IOM is managed in the OR. 

- was there any protocol to keep MAPs>80 intraoperatively? 

- are patients with IOM changes getting further imaging? what if it was due to low blood flow and spinal infarcts? any attempts of diagnosing the reason of the IOM? different prognosis may arise from different etiology. 

no need to change. 

Author Response

POINT 1: did any of the patient received instrumentation? if so, how many levels? receiving instrumentation is crucial to adjust for. if no instrumentation was used, I encourage the authors to add it to the methods. 

RESPONSE 1: We thank the Reviewer for having pointed at this issue. No instrumentation was used. We inserted the following sentence in the text, in the Material and Methods section, lines 106-108: "all the patients underwent laminotomy and subsequent laminoplasty, sparing the articolar joints in order to preserve the spinal stability and avoid instrumentation". 

POINT 2:  how many patients achieved total resection? what was the operative time? 

RESPONSE 2: We added the following sentence in the text, Results section, lines 175-179: "In all the patients a gross total resection was obtained, as assesed by postoperative MRI, routinely performed within 48-72 hours after surgery. In all 19 cases of meningioma a Simpson grade II resection was obtained, with dural preservation thus allowing an easier layer reconstruction. The mean surgical time was 3.19 ±1.13 hours".

POINT 3: How was a change in IOM approached? is there any protocol at the hospital or a list to run regarding things to check that may contribute to the changes, such as patient positioning?

RESPONSE 3:  We thank the Reviewer for having given us the opportunity to detail our protocol for management of IONM changes. We inserted the following sentences in the text, Material and methods section, lines 99-100: "IONM were recorded during the whole procedure from patient positioning to laminoplasty and wound closure". Lines 103-106: "In case of modifications of IONM data, the first due check is related to any changes in anaesthesiologic plan: every administration of curare or anesthetic gas requires an adequate disposal period. We usually acquire IONM data before and after  pronating the patients, in order to identify any changes and, in case, adjust the position". Lines 109-112: "IONM changes can occur durging surgical dissection: in these cases the surgeon usually proceeds to irrigation with saline solution, modifications in surgical strategy and, in some cases, systemic administration of steroids".

POINT 4: was a power analysis done? I would emphasize on that in the methods, as the results are clearly underpowered, especially the KM plots. 

RESPONSE 4: The a priori power analysis has been added in the Methods section, lines 155-160 and 164-167. The power analysis showed a remarkable 97.5% power for the comparison of MEP amplitude ratios between patients with good vs poor 1-year follow-up, though a normal distribution of the amplitude ratio values had to be postulated for the calculation to be performed. Instead, the power of the ROC curve, by postulating an ideal area under the curve of 0.8, was indeed suboptimal (65%). To note, we had already stated that the ROC analysis led to suboptimal results (see page 6, line 230).

POINT 5: Please tone down your conclusions and remove "clear evidence"

RESPONSE 5: we agree with the Reviewer's suggestion: the conclusion was revised and "clear evidence" was removed. 

POINT 6: please discuss the limitations in more depth, including the low number of patients involved. 

RESPONSE 6: the limitations were revised and better expicitated, highlighting the low number of included patients.

POINT 7: how many surgeons were involved? 

RESPONSE 7: This issue was explicitated in the Material and methods section, lines 108-109, where the following sentence was added: "The procedures were performed by four surgeons of the same equipe".

POINT 8: what was the approach used? are they all posterior? was there any costotransversectomies? 

RESPONSE 8: As stated in Response 1, all the patients underwent laminotomy and subsequent laminoplasty, sparing the articolar joints.

POINT 9: Please add to the discussion a paragraph about how, at your institution, IOM is managed in the OR. 

RESPONSE 9: We inserted the following sentence in the text, Discussion section, lines 282-287: "At our institution IONM is currently applied during each surgical procedure, either in cranial, spinal or peripheral districts, at risk for neurological functions impairment. The large availability of this precious tool in our experience is mainly due to organizational issues: IONM service (either devices and technical and medical staff) relates directly to neurosurgical department. Routinely application of IONM allowed us to increase experience and to reduce OR setup time".  

POINT 10: was there any protocol to keep MAPs>80 intraoperatively?

RESPONSE 10: Yes, we have a specific protocol about intraoperative MAPs. This point was added to Material and methods section, lines 100-103.  

POINT 11: are patients with IOM changes getting further imaging? what if it was due to low blood flow and spinal infarcts? any attempts of diagnosing the reason of the IOM? different prognosis may arise from different etiology. 

RESPONSE 11: as outlined in Response 2, each patient underwent MRI within 48-72 hours after surgery. In some cases MRI revealed a modification of imaging related to myelomalacia, hovever imaging didn't allow to recognize the specific etiology of IONM changes.

Reviewer 2 Report

The manuscript reported the use of IONM in patients with intradural extramedullary spinal tumors. The comments are:

1. In Table 2, the percentage for poor MMS is 21.7%, not 1.7%.

2. The percentage of schwannomas in this study is higher than other reports. The location of meningiomas (dorsal vs ventral), and schwannomas (cervical, thoracic, lumbar) should be included in the results.

3. The size of tumor usually affects the extent of spinal cord/nerve root compression, and surgical risks. This factor should be included in the analysis. 

4. Were there patients whose MEP was not detectable before the surgery? The results of this were not clear.

5. Abbreviations should be provided below each table.

6. A few figures showing the cases with worsened SEP or MEP after surgery will be important.

Author Response

POINT 1:  In Table 2, the percentage for poor MMS is 21.7%, not 1.7%.

RESPONSE 1: we corrected the typing error. We apologize for that.

POINT 2: The percentage of schwannomas in this study is higher than other reports. The location of meningiomas (dorsal vs ventral), and schwannomas (cervical, thoracic, lumbar) should be included in the results.

RESPONSE 2: Thank you for your comment: we revised our data and we added these points in the Results section (lines 171-175).

POINT 3: The size of tumor usually affects the extent of spinal cord/nerve root compression, and surgical risks. This factor should be included in the analysis. 

RESPONSE 3: we categorized the tumors in smaller than 2 cm, between 2 and 5 cm and larger than 5 cm. There were no differences in terms of outcome between the first two groups; conversely tumours larger than 5 cm were negatively associated with the outcome at discharge and at final follow-up. We added these data in the Results section (lines 206-210, 222-223). 

POINT 4:  Were there patients whose MEP was not detectable before the surgery? The results of this were not clear.

RESPONSE 4: there weren't patients whose MEPs were not detectable before surgery, although in some of these an impaired motor function was detected, as outlined in Table 2. We clarified this point in the Results section (lines 183-186)

POINT 5: Abbreviations should be provided below each table.

RESPONSE 5: we appreciated the observation: explanation of abbreviations were inserted below each table.

POINT 6: A few figures showing the cases with worsened SEP or MEP after surgery will be important.

RESPONSE 6: We added a novel figure (Figure 1) showing the worsening of IONM in a case of thoracic meningioma surgery.

Round 2

Reviewer 1 Report

the authors responded to all comments and addressed the relevant changes. Approve. 

Reviewer 2 Report

I have checked the response from the authors and the revised manuscript. I think the manuscript has been revised according to my suggestions.